# Machine Learning Identification of Obstructive Sleep Apnea Severity through the Patient Clinical Features: A Retrospective Study

**DOI:** 10.3390/life13030702

**Published:** 2023-03-05

**Authors:** Antonino Maniaci, Paolo Marco Riela, Giannicola Iannella, Jerome Rene Lechien, Ignazio La Mantia, Marco De Vincentiis, Giovanni Cammaroto, Christian Calvo-Henriquez, Milena Di Luca, Carlos Chiesa Estomba, Alberto Maria Saibene, Isabella Pollicina, Giovanna Stilo, Paola Di Mauro, Angelo Cannavicci, Rodolfo Lugo, Giuseppe Magliulo, Antonio Greco, Annalisa Pace, Giuseppe Meccariello, Salvatore Cocuzza, Claudio Vicini

**Affiliations:** 1Department of Medical and Surgical Sciences and Advanced Technologies “GF Ingrassia” ENT Section, University of Catania, 95123 Catania, Italy; 2Sleep Surgery Study Group of the Young-Otolaryngologists of the International Federations of Oto-rhino-laryngological Societies (YO-IFOS), 75001 Paris, France; 3Department of Mathematics and Informatics, University of Catania, 95123 Catania, Italy; 4Otorhinolaryngology Department, Sapienza University of Rome, Policlinico Umberto I, Viale del Policlinico 151, 00010 Rome, Italy; 5Department of Human Anatomy and Experimental Oncology, Faculty of Medicine, UMONS Research Institute for Health Sciences and Technology, University of Mons (UMons), 7000 Mons, Belgium; 6Department of Otorhinolaryngology and Head and Neck Surgery, Foch Hospital, School of Medicine, UFR Simone Veil, Université Versailles Saint-Quentin-en-Yvelines (Paris Saclay University), 75001 Paris, France; 7Department of Head-Neck Surgery, Otolaryngology, Head-Neck, and Oral Surgery Unit, Morgagni Pierantoni Hospital, 47121 Forlì, Italy; 8Service of Otolaryngology, Rhinology Unit, Hospital Complex of Santiago de Compostela, 15701 Santiago de Compostela, Spain; 9Department of Otorhinolaryngology-Head and Neck Surgery, Hospital Universitario Donostia, 20001 San Sebastian, Spain; 10Otolaryngology Unit Santi Paolo e Carlo, Hospital Department of Health Sciences, Università Degli Studi di Milano, 20021 Milan, Italy; 11Department of Otorhinolaryngology, Grupo Medico San Pedro, Monterrey 64660, Mexico; 12Department ENT and Audiology, University of Ferrara, 44121 Ferrara, Italy

**Keywords:** obstructive sleep apnea, artificial intelligence, machine learning, OSA severity, ESS, clinical OSA scores

## Abstract

Objectives: To evaluate the role of clinical scores assessing the risk of disease severity in patients with clinical suspicion of obstructive sleep apnea syndrome (OSA). The hypothesis was tested by applying artificial intelligence (AI) to demonstrate its effectiveness in distinguishing between mild–moderate OSA and severe OSA risk. Methods: A support vector machine model (SVM) was developed from the samples included in the analysis (N = 498), and they were split into 75% for training (N = 373) with the remaining for testing (N = 125). Two diagnostic thresholds were selected for OSA severity: mild to moderate (apnea–hypopnea index (AHI) ≥ 5 events/h and AHI < 30 events/h) and severe (AHI ≥ 30 events/h). The algorithms were trained and tested to predict OSA patient severity. Results: The sensitivity and specificity for the SVM model were 0.93 and 0.80 with an accuracy of 0.86; instead, the logistic regression full mode reported a value of 0.74 and 0.63, respectively, with an accuracy of 0.68. After backward stepwise elimination for features selection, the reduced logistic regression model demonstrated a sensitivity and specificity of 0.79 and 0.56, respectively, and an accuracy of 0.67. Conclusion: Artificial intelligence could be applied to patients with symptoms related to OSA to identify individuals with a severe OSA risk with clinical-based algorithms in the OSA framework.

## 1. Introduction

Obstructive sleep apnea (OSA) is a respiratory disorder characterized by the partial or total collapse of the upper airways with intermittent hypoxia, a chronic systemic inflammatory state, and an increased cardiovascular risk [1]. OSA is also associated with comorbidities, such as metabolic syndrome and olfactory or neurodegenerative disorders. Although OSA has a very high prevalence worldwide, up to 50% of the general population, subjects are often unaware of the disorder, and the diagnosis is performed when the associated comorbidities have already developed [2].

The strong connection between the severity of obstructive apnea identified through standard diagnostic tests, such as nocturnal polysomnography or pulse oximetry, and the associated cardiovascular or neurodegenerative risks are established in the literature [3,4]. Conversely, the different clinical features and scoring systems, such as age, BMI, the anatomical scores of palatal collapses, or validated questionnaires, although significantly correlating with pathology, do not possess adequate sensitivity or specificity to substitute instrumental diagnostics tools [5].

Recently, Sutherland et al. analyzed in a patient-level meta-analysis the relationship between craniofacial morphology and weight loss with sleep apnea severity [6]. The authors demonstrated a correlation between weight and AHI changes (rs = 0.3, *p* = 0.002) and an increased maxilla–mandible relationship angle related to AHI improvement (β [95% CI] −1.7 [−2.9, −0.5], *p* = 0.004) at linear regression.

In this regard, clinical prediction decision-making algorithms that use clinical information, objective scores, and easy-to-execute subjective questionnaires could be useful strategies for predicting OSA [7].

Maranate et al. in 2015 proposed a prioritization process of clinical risk factors of the severity of OSA using the analytical hierarchy process to select the patients with the greatest need for an in-depth diagnostic study [8]. The authors developed an algorithm used on 1042 suspected OSA patients who had undergone diagnostic PSG study. Moreover, 42 variables of disease severity were identified, with an overall sensitivity/specificity of the model for severe, moderate, and mild of 92.32%/91.76%, 89.52%/88.18%, and 91.08%/84.58%, respectively.

Artificial intelligence (AI) is rapidly gaining importance in medicine, opening promising new perspectives [6,7,8]. Through a deep learning system, different types of data can be exploited, such as clinical scores, questionnaires, or diagnostics [2,4,5].

Tsuiki et al. in 2021 developed a deep convolutional neural network (DCNN) using data from lateral cephalometric radiographs of 1389 subjects and tested on 10% of the enrolled sample (*n* = 131) [7]. The DCNN, as well as manual cephalometric analyses, significantly predicted the presence of severe OSA, with a full image sensitivity/specificity of 0.87/0.82 (χ^2^ = 62.5, *p* <0.01), demonstrating promising prospects for AI in the triage of OSAS. Nevertheless, these forecasting tools’ predictive performances and features differ significantly between literature studies, limiting their generalizability and practicality.

The aim of our study was to evaluate the usefulness of the application of the SVM algorithm predicting the severity of OSA through clinical parameters, subjective questionnaires, anatomical scores, and associated comorbidities.

## 2. Materials and Methods

### 2.1. Study Design and Data Collection

Guidelines on strengthening the reporting of observational studies in epidemiology (STROBE) were followed [9]. We carried out a multicentric retrospective study conducted at the Ear, Nose, and Throat Unit (ENT) of our hospitals from 1 January 2010 to 31 December 2021 (Figure 1).

Participants aged ≥ 18 years who were referred to our units for sleep respiratory disorders were enrolled and subjected to clinical diagnostic evaluation and consequently phenotyped. A full clinical history of symptoms, such as morning headache and decreased libido, validated questionnaires (ESS), and anatomical or endoscopic scores were collected (Table 1) (Appendix A) [10,11,12,13,14,15,16,17,18,19,20,21].

All the participants underwent a sleep study that was carried out in an unattended way by means of a Polymesam Unattended Device 8-channel and then reviewed and scored by the same expert in sleep medicine according to the American Academy of Sleep Medicine (AASM) Guidelines [3].

We selected two diagnostic thresholds for OSA: mild to moderate (AHI ≥ 5 events/h and AHI < 30 events/h) and severe (AHI ≥ 30 events/h), according to the latest AASM guidelines on OSA management [3]. The algorithm was trained and tested using the two established threshold parameters to predict OSA patients’ severity.

### 2.2. Statistical Analysis

Standard descriptive statistics were used, reporting mean and standard deviation for continuous variables and percentages for categorical ones. The independent t-test was performed for normally distributed values, while the Mann–Whitney U test was used for abnormally distributed values. The chi-square test was performed to test the observed and expected data differences. A value of *p* < 0.05 was deemed to be statistically significant. All analyses were performed using the Social Sciences Statistical Program (IBM SPSS Statistics for Windows, IBM Corp. Released 2017, Version 25.0 Armonk, NY, USA: IBM Corp).

### 2.3. Stratification Process 

To perform the stratification of participants, we selected among independent variables assessed for disease severity the AHI cutoff according to the AASM guidelines to define disease severity. Thus, we stratified patients into a mild–moderate OSA group when the AHI value was ≤30; instead, a severe OSA group was defined when a value of AHI > 30 was found. Other variables identified were instead introduced into the predictive models as independent variables and converted into a binary value according to the respective cutoff definition.

Consequently, the splitting process of training and testing data divided the sample (N = 498) into two different homogeneous datasets. The training group included 75% (N = 373) of participants while the remaining 25% (N = 125) were used for testing. The performance of the logistic regression and SVM classifier models was tested according to the two AHI thresholds mild/moderate (AHI ≤ 30) and severe (AHI > 30).

The study protocol was approved by the Ethical Committee and was conducted in accordance with the declaration of Helsinki. 

We explored the data to improve their quality, handling missing or removing duplicate values, managing the existence of outliers or anomalies (data points differing substantially from the rest of the data), converting invalid or bad formatted values.

### 2.4. Logistic Regression Model

Logistic regression predictive models were used to classify patients and evaluate the performance of predictors. We used receiver operating characteristic (ROC) curves to assess the ability of the logistic regression models to identify patients with mild/moderate or severe AHI. Results were reported in terms of area under the curve (AUC) and 95% confidence interval (95% CI). A first multivariate logistic regression model was used to evaluate the model accuracy in an outcome prediction with the complete set of variables (full model). A second multivariate logistic model was the result of a backward stepwise elimination for selecting features and eliminating the ones that did not have a significant effect on the dependent variable or prediction of outcome to find a reduced model that best explained the data. We first worked on patients with a complete assessment of the following information: Age, gender, BMI, familiarity with OSAS, hypertension, cardiovascular disorders, diabetes, dyslipidemia, COPD, anxiety/depression, septoturbinoplasty, tonsillectomy, snoring, choking, morning headache, decreased libido, ESS, septal deviation, internal valve collapse, external valve collapse, lower turbinate hypertrophy, adenoid hypertrophy, Friedman tonsils score, Mallampati score, Friedman palate score, palate phenotype according to Woodson classification, endoscopic lingual tonsils score, retropalatial Mueller maneuver, retrolingual Mueller maneuver, panting test, retrognathia, and upper jaw contraction. All tests were performed at a significance level α = 0.05. After backward elimination, the reduced model included the following features: Age, gender, BMI, diabetes, anxiety/depression, choking, and septal deviation.

We consequently evaluated the multicollinearity of the logistic regression model due to the different features included. Therefore, a stepwise approach was performed in which previously removed features were individually reintroduced into the model to assess the risk of overfitting and the influence of individual values in the model.

### 2.5. SVM Model

The same training/test set and features from the full logistic regression were later used to develop the SVM model. Several datasets are not linearly separable even in a feature space, not satisfying all the constraints in the minimization problem of SVM. To fill this gap, Slack variables are introduced to allow certain constraints to be violated. By choosing very large slack variable values, we could find a degenerate solution that would lead to model overfitting [22]. To penalize the assignment of too large slack variables, the penalty is introduced in the classification objective:ε_i_, indicates slack variables, one for each datapoint i, to allow certain constraints to be violated.C, indicates a tuning parameter that controls the trade-off between the penalty of slack variables ε_i_ and the optimization of the margin. High values of C penalize slack variables leading to a hard margin, whereas low values of C lead to a soft margin, which is a bigger corridor that allows certain training points inside at the expense of misclassifying some of them. In particular, the C parameter sets the confidence interval range of the learning model.

The radial basis fFunction (RBF) kernel function expression on two samples, x∧x^′, is defined as K (x,x^′) = exp(-γ|(|x-x^′|)|^2) where |(|x-x^′|)|^2 is the squared Euclidean distance between the two feature vectors, and γ is a free parameter. The RBF can be applied to a dataset by choosing two parameters, C and γ. The classifier performance of SVM depends on the choice of these two parameters. 

A grid search method was used to find the optimal parameters of the RBF for SVM. This method considered m values in C and n values in γ, according to the M × N combination of C and γ, by training different SVMs using K-fold cross-validation. Here, we used a grid search on a 5-fold cross-validation to optimize accuracy. Thus, we selected as optimal parameters for the RBF kernel in the SVM model a γ value of 0.5 and C of 100.

Consequently, a Shapley plot (SHAP) was calculated. SHAP, which stands for Shapley Additive exPlanations, is an interpretability method based on Shapley values, a solution concept in cooperative game theory named in honor of Lloyd Shapley, who introduced it in 1951. SHAP was consequently applied to explain individual predictions of any machine learning (ML) model. The explanation model is represented by a linear model—an additive feature attribution method—or just the summation of present features in the coalition game. After the generation of Shapley plots, each predictor contribution to the SVM model output was described in terms of the SHAP average value. 

The independent variables were thus ranked in descending order of importance. Instead, the violin plot generated shows the impact of a value associated with higher or lower prediction and positive or negative correlation on the X-axis. The color correlates with the average feature value at the plot position: Red areas represent highly valued features while blue areas are low. The violin plot also shows the outliers drawn as points.

The analyses were conducted using Python 3.6.9 with Statsmodel 0.10.2, Scikit-learn 1.0.2, and Shap 0.40 libraries.

Shapley plots show the contribution of each predictor to the SVM model output in terms of SHAP value. The variables were ranked by importance in descending order and the color represents the average feature value at that position. The violin plot shows the average medium SHAP value of each independent variable; the violin plot shows the impact of a value associated with higher or lower prediction and positive or negative correlation. The violin plot also shows the outliers drawn as points.

### 2.6. Models Test Analysis 

Comparison of the accuracy of the diagnostic tests performed was analyzed by an analysis of the areas under the receiver operating characteristic (ROC) curves. Models with the same characteristics and the same training/test set were analyzed through a Z-test to evaluate a statistical difference. The Z-test is a parametric statistical test used to evaluate whether the mean of a given distribution differs significantly from a hypothesized value.

### 2.7. Reporting Completeness of Machine Learning Study

We evaluated the reporting completeness of this research study referring to the TRIPOD (transparent reporting of a multivariable prediction model for individual prognosis or diagnosis; www.tripod-statement.org, accessed on 8 February 2023) checklist for prediction model validation (accessed on 6 February 2023).

This statement contains a 20-item checklist, for a total of 31 items, with all sub-items included. The checklist contains questions about the title, abstract, background, methods, results, discussion, Appendix A, and funding information. 

Each included item received a score of “1” for adherence and a score of “0” for non-adherence. Multiple items (items 1, 2, 3a, 4b, 5a, 6a, 7a, 7b, 9, 10a, 10b, 10d, 13a, 13b, 14a, 15a, 16, 17, 20, and 22) in the TRIPOD analysis were derived from several sub-items (the sub-items for each number can be found in www.tripod-statement.org (accessed on 7 June 2021)). The results of each TRIPOD item for each paper and the level of reporting adherence for each TRIPOD item were documented systematically in a spreadsheet.

We thus obtained a TRIPOD adherence score of 93,45% (29/31 items) by dividing the sum of TRIPOD items adhered to by the entire number of applicable TRIPOD items in the study.

## 3. Results

### 3.1. Patients Features 

After selection, a total of 498 participants, with an age of 50.96 ± 12.15 years, were included in the study, of which 427/498 (87.76%) were male vs. 61/498 (12.24%) female (Table 1). The mean BMI was 27.32 ± 4.02 kg/cm. A mean AHI value of 37.21 ± 23.24 events/h and a mean oxygen desaturation index (ODI) of 35.37 ± 24.79 events/h were reported. An overall 220/498 (44.17%) of the participants were identified as mild to moderate OSA (AHI < 30 events/h), while 278/498 (55.82%) of the participants as severe OSA (AHI threshold ≥ 30 events/h). 

Subsequently, the participants were divided into two homogeneous training (N = 373) and test (N = 125) groups, homogeneous for the independent variables included in the analysis (Figure 1).

### 3.2. Logistic Regression Analysis, Full and Reduced Models

Through the traditional statistical analysis, the full model demonstrated a ROC curve with an AUC of 0.73 (95% CI = 0.64–0.82) and 0.68 accuracy (Figure 2a). 

The sensitivity and specificity of the regression to distinguish among OSA severity were, respectively, 0.74/0.63. 

The consequent features selection using the backward stepwise elimination demonstrated a reduced logistic regression model with a sensitivity/specificity of 0.79/0.56, an accuracy of 0.67, and an AUC of 0.69 (95% CI = 0.6–0.78) (Figure 2b).

The assessment of the multicollinearity of the model by re-introducing the removed features one-by-one did not demonstrate a statistically significant difference in model performance.

### 3.3. SVM Model Performance and ROC Curve Analysis 

The SVM algorithm was adopted to discriminate OSA severity among participants. Through the algorithm, all the overall performance scores as accuracy (number of correct predictions/total number of predictions), ROC AUC (area under the curve), sensitivity (true positive rate), and specificity (true negative rate) were improved.

In particular, the SVM model demonstrated a recall and precision score of 0.80 and 0.93, respectively, to assess mild to moderate OSA; instead, the outcomes were 0.93 and 0.81 to identify severe OSA (Table 2). 

To avoid bias within the dataset, we evaluated the model score using a 10-fold stratified cross-validation strategy, obtaining an average accuracy score of 0.87 (95% CI = [0.83, 0.91]). The result of the stratified cross-validation is in line with the accuracy score of the model trained on 75% of the dataset and tested on the remaining 25%. The algorithm provided a ROC with an AUC of 0.92 (95% confidence interval = 0.87–0.97) (Figure 3). 

The clinical variables included in the model and the Shapley plot consequently generated showed the contribution of each predictor to the SVM model, as summarized in Figure 4a,b. 

Among the independent variable predictors of OSAS severity assessed, dyslipidemia (0.0623), choking (0.0588), diabetes (0.0576), mood disorders (0.0486), and familiarity for OSAS (0.0452) demonstrated a higher average impact on model output magnitude in terms of an absolute mean SHAP value. The ROC AUC comparison between the complete logistic regression model (AUC = 0.73) and the SVM model (AUC = 0.92) through the Z-test confirmed a statistically significant difference among the two models (*p* < 0 0.001). 

Instead, the ROC AUC comparison between full and reduced logistic regression model was not significant (*p* = 0.541).

## 4. Discussion

Our study demonstrated how artificial intelligence can be useful in assessing patients with OSA-related symptoms and determining the risk of disease severity with clinic-based algorithms. We reported a sensitivity and specificity significantly higher for the SVM model than classical logistic regression. Indeed, as reported in our results through the SVM model, it is possible to stratify OSA patients according to the severity with a higher accuracy of 0.86 compared with the full logistic regression mode accuracy of 0.68.

### 4.1. Diagnostic and Therapeutic Role

The use of AI technology in clinical practice is an emerging and debated topic, both for its possible diagnostic and therapeutic implications. Through ML, it is possible to exploit the multiple variables of easy and rapid extraction, such as a patient’s clinical history and anthropometric or demographic characteristics, to facilitate the identification of otherwise complex pathologies [6]. 

The field of breathing sleep disorders could also benefit from improving ML Technology, using both its application in early diagnosis and the analysis of predictive factors of response to medical or surgical treatment [7,8,23].

Kim et al. in 2021, through a preoperative machine learning-based clinical mode, improved the prediction of the therapeutic sleep surgery outcome with higher accuracy for the gradient boosting model than the logistic models [24]. 

The application of the algorithm elaborating several parameters collected at baseline visits between adherent and non-adherent groups demonstrated a sensitivity of 68.6% and an AUC of 72.9% through the vector machine model.

### 4.2. Diagnostic Application of AI

Sleep breathing disorders lend themselves well to the diagnostic application of AI as they are often strongly correlated with elevated BMI, and cardiovascular, metabolic, or central nervous system disorders.

Holfinger et al. assessed the diagnostic performance of OSA machine learning prediction tools using readily available data, such as age, sex, BMI, and race, and compared the efficacy with a STOP-BANG-based model [23]. The authors included a wide cohort of 17,448 subjects in a retrospective study, demonstrating that AUCs (95% CI) of the kernel support vector machine (0.66 [0.65–0.67]) were significantly higher than logistic regression ones (0.61 [0.60–0.62]).

Machine learning-derived algorithms may also improve and simplify the widespread identification of OSA in the pediatric population, providing better diagnostic performance than logistic regression with patient-reported symptoms [8,24,25,26]. 

Gutiérrez-Tobal et al. performed a systematic review of studies assessing the reliability of a machine learning-based method implementation of OSA detection in clinical practice [25]. The authors retrieved 90 studies involving 4767 different pediatric subjects and demonstrated an improved ML diagnostic performance on OSA severity criteria (sensitivity = 0.652; specificity = 0.931; and AUC = 0.940). 

However, an important aspect that has not been analyzed on the advantages of artificial intelligence in OSA is its possible use in identifying OSA pathology concerning the healthy population, determining which patients are at risk, and optimizing the use of diagnostic resources, such as polysomnography [6,7].

Our analysis confirmed the superiority of vector models using SVM models in determining disease severity in patients with OSA compared to traditional logistic regression models. In fact, we found a significantly greater sensitivity/specificity of 0.93/0.80, and an accuracy of 0.86 for the SVM than the logistic regression full mode.

It is well-known in the literature that OSA is associated with obesity and cardiovascular and cerebrovascular disorders. Several mechanisms occur in patients with OSA, including a chronic inflammatory state, intermittent hypoxia, and even alterations of their lipid profile, probably due to a reduction in androgen levels. Moreover, OSA severity and lipid-related comorbidities, such as atherosclerosis, possess a well-known correlation.

### 4.3. OSA Risk Factors and Comorbidities

Our SVM model demonstrated dyslipidemia as the highest average SHAP feature value (0.0623) for OSA severity among clinical variables included in the diagnostic algorithm. Conversely, its predictive role was not confirmed in the logistic regression full model analysis (*p* = 0.123). Although OSA is commonly associated with craniofacial anomalies and palatal or base tongue disorders validated as upper airway obstruction sites by DISE, no research study has evaluated the predictive role of such features using a machine learning algorithm. Craniofacial variables represent noteworthy risk factors for OSA. The risk of OSA in adult subjects with altered craniofacial anatomy on lateral cephalograms is well-known in the literature [6]. Among the variables with significant heterogeneity, the position and length of the mandible (BNS: −1.49° and Go-Me: −5.66 mm), the area of the tongue (T: 366.51 mm^2^), and the soft palate (UV: 125, 02 mm^2^), and upper airway length (UAL: 5.39 mm) were identified as strongly correlating with the presence of OSA. Our SVM analysis demonstrated a higher average SHAP feature value for the palate phenotype (0.0308), Friedman tonsils score (0.0363), and palate score (0.0623) for OSA severity than other clinical features assessed in the model. However, the predictive role in the logistic regression full model analysis was not significant for the palate phenotype (*p* = 0.148), while Friedman’s tonsils score (*p* = 0.041) and Friedman’s palate score (*p* = 0.038) were statistically significant. 

A systematic review of the prevalence of OSA in an Asian population reported interesting data on highly related features, such as gender, older age, BMI increase, and arterial hypertension, that significantly correlated with the onset of sleep breathing disorders and severity [4]. The difference in gender prevalence rates was also supported by our analysis, with gender, age, and hypertension showing a SHAP value of 0.0435, 0.0336, and 0.0408, respectively. Conversely, in the logistic regression full model, the predictive role in the analysis was significant for age (*p* = 0.017) and gender (*p* = 0.019), while not for hypertension (*p* = 0.649). 

### 4.4. Study’s Limitations

Prediction tools that determine OSA risk include variables, such as patient-reported symptoms through symptom questionnaires; however, there are often unavailable or inaccurate data from large groups of individuals with multicenter studies using electronic medical records that are not similar across institutions.

Although the machine learning model demonstrated significant AUC and adequate sensitivity and specificity, our study’s main shortcoming was the reduced sample with which the model was trained. In fact, it is known how artificial intelligence models become proportionally more effective as new subjects are added, with a refinement of the predictive algorithm. A consequence of this limitation is that in our model, highly significant features in the literature, such as BMI, Mueller maneuver, ESS, or cardiovascular disorders, do not take on a dimension in the model that reflects clinical severity. 

Ultimately, although the verification of inputs and data values is important, we must also consider the validation of the data model itself. In fact, our data model could be not correctly structured, leading to several biases. Therefore, a future perspective to enable the potential clinical application of our model will be to validate it using independent data.

## 5. Conclusions

The development of an AI model to predict the risk of developing OSA severity has shown promising prospects for application in a clinical setting after adequate training and sufficiently large samples. Using demographic data associated with easily detectable clinical or endoscopic scores, a practical model for predicting OSA severity in the future could be possible.

## Figures and Tables

**Figure 1 life-13-00702-f001:**
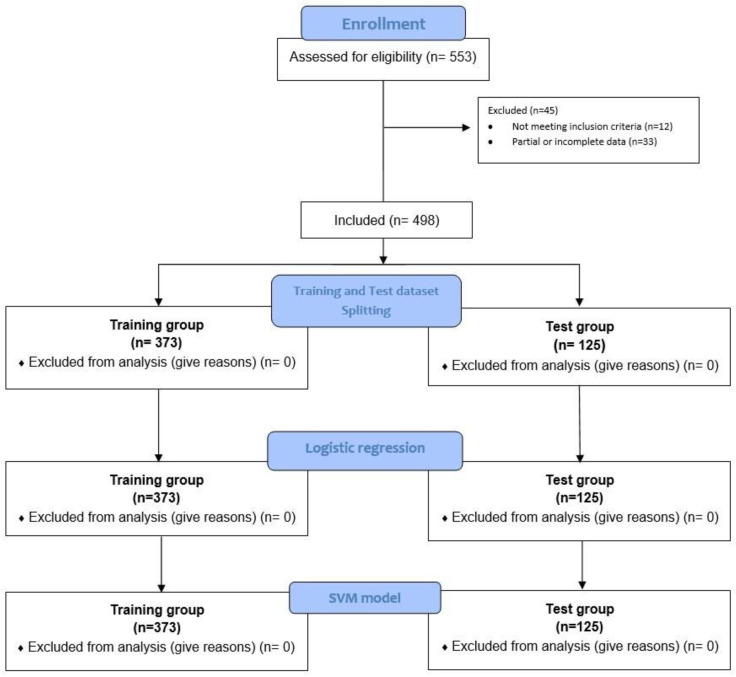
Flow diagram of preprocessing, dataset splitting, and SVM model training.

**Figure 2 life-13-00702-f002:**
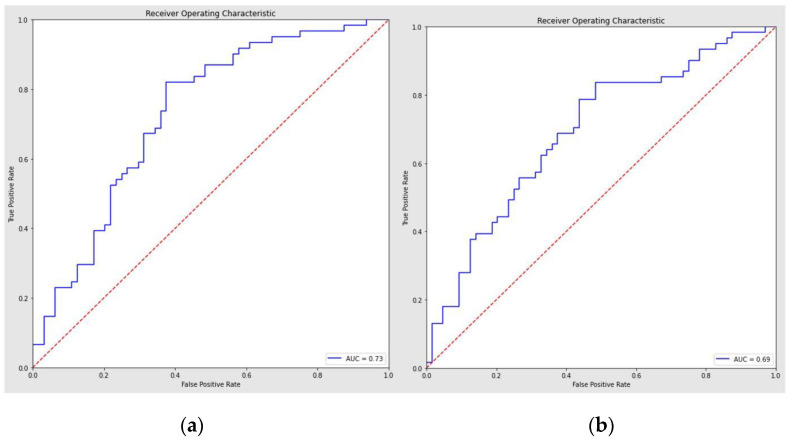
(**a**) Logistic regression full model. (**b**) Logistic regression classifier reduced model (features selection after backward elimination). List of selected features: Age, gender, BMI, diabetes, anxiety/depression, choking, septal deviation.

**Figure 3 life-13-00702-f003:**
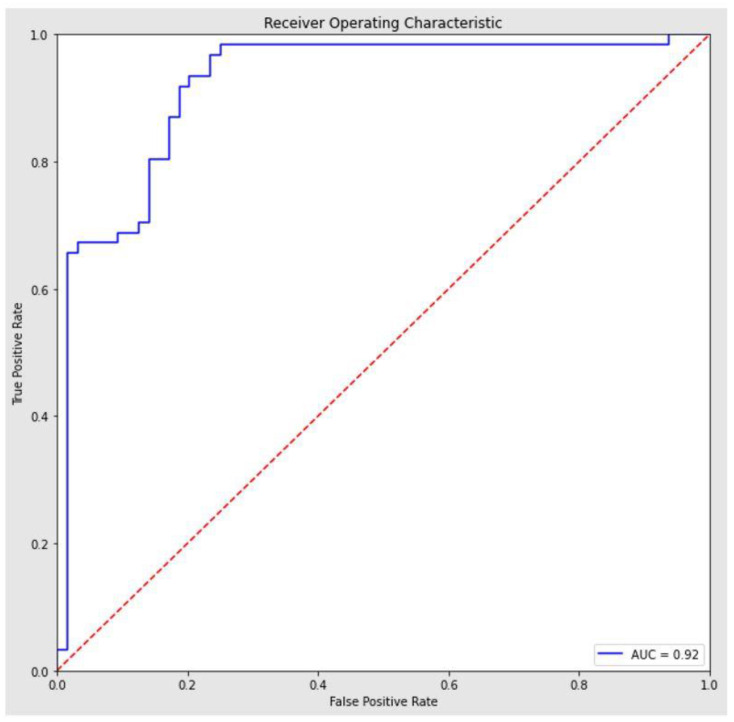
ROC SVM model.

**Figure 4 life-13-00702-f004:**
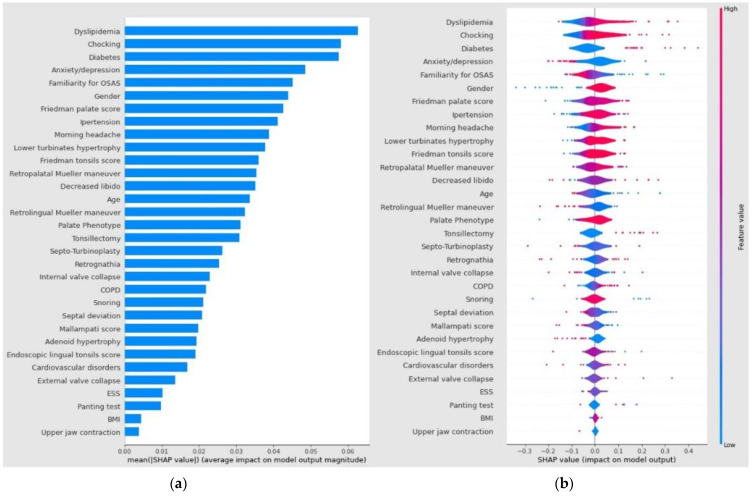
Shapley plots show the contribution of each predictor to the SVM model output in terms of SHAP value. The variables were ranked by importance in descending order, and the color represents the average feature value at that position. (**a**) The violin plot shows the average medium SHAP value of each independent variable; (**b**) the violin plot shows the impact of a value associated with higher or lower prediction and positive or negative correlation. The violin plot also shows the outliers drawn as points.

**Table 1 life-13-00702-t001:** Main demographic features.

Characteristics	Total (*n* = 498)	Mild–Moderate OSA (*n* = 220)	Severe OSA (*n* = 278)	*p*-Value
**Age**	50.96 ± 12.15	51.57 ± 12.03	50.47 ± 12.20	*0.315*
**Gender**				
male	427/498 (87.76%)	179/498 (35.94%)	248/498 (49.79%)	*0.189*
female	61/498 (12.24%)	31/498 (6.22%)	30/498 (6.02%)	
**AHI**	37.21 ± 23.24	17.84 ± 7.50	53.96 ± 17.56	*<0.001*
**ODI**	35.37 ± 24.79	17.76 ± 17.82	49.38 ± 20.13	*<0.001*
**Mean SpO2**	92.33 ± 3.07	93.35 ± 2.25	91.53 ± 3.38	*<0.001*
**Lower SpO2**	75.92 ± 12.13	80.05 ± 11.87	72.65 ± 11.31	*<0.001*
**BMI**	27.32 ± 4.02	26.38 ± 2.74	28.06 ± 4.66	*<0.001*
**ESS**	7.97 ± 4.92	7.26 ± 4.43	8.54 ± 5.20	*0.003*

**Table 2 life-13-00702-t002:** SVM model classification report. Accuracy, number of correct predictions/total number of predictions; ROC AUC, area under the curve; sensitivity, true positive rate; specificity, true negative rate. ^a^ SVM ROC AUC vs. Full regression model. ^b^ SVM ROC AUC vs. Reduced regression model. ^c^ Full regression ROC AUC vs. reduced regression.

Model	AHI	Precision	Recall	F1-Score	Sensitivity	Specificity	Accuracy	*p*-Value
Full Logistic	mild–moderate	0.71	0.62	0.67				
	severe	0.65	0.74	0.69	*0.74*	*0.63*	*0.68*	*p < 0.001 ^a^*
Reduced Logistic	mild–moderate	0.73	0.56	0.64				
	severe	0.63	0.79	0.70	*0.79*	*0.56*	*0.67*	*p < 0.001 ^b^*
SVM	mild–moderate	0.93	0.80	0.86				
	severe	0.81	0.93	0.87	*0.93*	*0.80*	*0.86*	*p = 0.541 ^c^*

## Data Availability

Data is unavailable due to privacy or ethical restrictions.

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
