# Peer review of "Machine Learning Identification of Obstructive Sleep Apnea Severity through the Patient Clinical Features: A Retrospective Study"

_life, 2023, doi:10.3390/life13030702_

Round 1

Reviewer 1 Report

In this interesting work, Doctor Maniaci and coauthors use artificial intelligence to identify among clinical scores potential risk factors of Obstructive Sleep Apnea severity. I really appreciated reading the paper and I think it is well-written. The rationale of the study is clear and well-presented. Methods and results are thoroughly described. I would like to rise only one point. The accuracy of the support vector machine model has been evaluated using a fixed training set composed by the 25% of the total dataset. To exclude any bias within the dataset it would be interesting to verify the accuracy using a stratified k-fold cross-validation strategy.

Author Response

thanks for the suggestions, here the response point by point

Reviewer 2 Report

This manuscript developed machine learning models, i.e., multivariable logistic regression model and SVM model for predicting the severity of OSA. The manuscript is well organized with clear writing. Some minor revisions are recommended:

·       The flow diagram in figure 1 can be improved: the logistic regression model and SVM model are parallel, not one after another.

·       There should be a citation about two diagnostic thresholds for OSA.

·       About 44% of the participants were identified as mild to moderate OSA. Is the split of the training and testing data also based on the severity of OSA?

·        Many paragraphs are just one sentence.

Author Response

dear reviewer, here the response point by  point.

Best regards

Reviewer 3 Report

This manuscript describes the development of a prediction model. While there could be clear clinical value in using such approach, I still have some comments regarding the paper.

As you are developing a prediction model, please check the TRIPOD checklist for prediction model development in addition to the STROBE guidelines.

How was data quality of the data guaranteed? As you are building models ,it is crucial to have high-quality data. Did any subject not meet the data quality requirements?

A lot of features were included in the full logistic regression model. This might have caused overfitting and p-values of the individual features might have been affected by including highly correlated features in the model. How did you assess multicollinearity of the model? Was a stepwise approach considered in which previously removed features could be included in the model again after removal of other (possibly correlated) features?

Please elaborate more on the SVM model in the methods section. This paragraph is currently hard to follow and lacks some overall information on the model used (included parameters, independent variable,…). Was the same training/test dataset used for logistic regression as for SVM? This is apparent from the figure but not from the text.

The Shapley plot is a part of the SVM analysis. Please clarify this in the text

It is stated in the results: “Subsequently, the participants were divided into two homogeneous training (N=373) and 273 test (N=125) groups, homogeneous for the independent variables included in the analysis 274 (Fig. 1).” Please elaborate on which independent variables were included in the stratification process in the methods section. It could be beneficial to add this as a separate subheading in the methods section.

Table II: What does the sensitivity, specificity and accuracy values represent in this table? Please clarify in the text and table heading.

Was the difference in AUC of the ROC curve statistically significant between both techniques? How did both models compare to each other?

How was the ‘palate phenotype’ defined? Was direction of collapse taken into account?

At this point, the discussion is quite hard to follow, please consider adding subheadings and restructuring the discussion to highlight the current findings and put them into perspective with previous research. Furthermore, please include a short summary of your results as the first paragraph in the discussion to guide the readers.

To allow future clinical application, your model should still be validated using independent data, please add this as a limitation/future perspective to your paper.

Author Response

Dear revisor, thanks for the suggestions. We think that the manuscript was significantly improved.

Best regards

Reviewer 4 Report

This is an interesting article written to demonstrate AI effectiveness in distinguishing patients with

mild-moderate OSA and severe OSA. The authors developed an algorithm to predict OSA severity

through the use of clinical parameters, subjective questionnaires and anatomical scores. The article

is generally well written and structured. The introduction provides sufficient background and

include all relevant references. However, I have a few observations:

1. conclusions section is missing in the abstract

2. improve the resolution of Figure 1

3. improve the structure of Table 1, the table is very confusing and full of data, arrange them

more clearly and legibly

4. line 183 put statistical analysis in italics

5. improve table structure 2, one row is offset

6. figure 4a and 4b improve resolution and magnification because is unreadable

Also, the quality of the writing could have been much better.

Author Response

Dear revisor, thank you for the suggestions, we think the manuscript was improved.

Best regards

Round 2

Reviewer 3 Report

I would like to thank the authors for their modifications, these have highly increased the quality of the manuscript. Please find some minor additional comments below:

 Please explicitly clarify the stratification variables. It is highly important for future understanding of the model to know which specific variables were used (eg. BMI cut-off, AHI cut-off?).

Please include some more information on the final model in the results section: final number parameters, finally selected parameters etc in the text or in a simple table comparing both final models. This would increase readability for a more clinical audience.

Author Response

Dear reviewer,

thanks for the suggestions. We attached the responses point by point.
